# Generation and Physiology of Hydrogen Sulfide and Reactive Sulfur Species in Bacteria

**DOI:** 10.3390/antiox11122487

**Published:** 2022-12-17

**Authors:** Sirui Han, Yingxi Li, Haichun Gao

**Affiliations:** Institute of Microbiology and College of Life Sciences, Zhejiang University, Hangzhou 310058, China

**Keywords:** reactive sulfur species, hydrogen sulfide, sulfur transformation, sensing and regulation

## Abstract

Sulfur is not only one of the most abundant elements on the Earth, but it is also essential to all living organisms. As life likely began and evolved in a hydrogen sulfide (H_2_S)-rich environment, sulfur metabolism represents an early form of energy generation via various reactions in prokaryotes and has driven the sulfur biogeochemical cycle since. It has long been known that H_2_S is toxic to cells at high concentrations, but now this gaseous molecule, at the physiological level, is recognized as a signaling molecule and a regulator of critical biological processes. Recently, many metabolites of H_2_S, collectively called reactive sulfur species (RSS), have been gradually appreciated as having similar or divergent regulatory roles compared with H_2_S in living organisms, especially mammals. In prokaryotes, even in bacteria, investigations into generation and physiology of RSS remain preliminary and an understanding of the relevant biological processes is still in its infancy. Despite this, recent and exciting advances in the fields are many. Here, we discuss abiotic and biotic generation of H_2_S/RSS, sulfur-transforming enzymes and their functioning mechanisms, and their physiological roles as well as the sensing and regulation of H_2_S/RSS.

## 1. Introduction

Sulfur, one of the most abundant elements on our planet, and an element that has six valence electrons, is essential to all living organisms. The term ‘reactive sulfur species (RSS)′, coined by Jacob and colleagues [1], has been in the general scientific vocabulary for more than two decades. Apparently, RSS, much like reactive oxygen species (ROS) and reactive nitrogen species (RNS), comprise chemically reactive molecules containing the element for which they are named, sulfur, oxygen, and nitrogen respectively [2]. Because of their highly reactive nature, the reactive species are prone to interact with cellular macromolecules, including nucleic acids, proteins, and lipids [3]. Not surprisingly, most, if not all, of these reactive species, all of which can be generated endogenously in the cell and arise abiotically in environments, are hostile to living cells at supraphysiological levels [2]. Despite this, in eukaryotes, especially mammals, a large majority of studies in fact center on the beneficial roles that these species play by mediating redox signaling and redox regulation [4]. It has been firmly established that these reactive species, as a signaling molecules, impact diverse processes including metabolism, vasodilation, neurotransmission, immunity, apoptosis, and cancer [2,5,6].

Compared to ROS and RNS, which have been extensively studied for decades, RSS are relatively new. The concept of RSS has evolved over time and is still far from unanimously held. The ‘narrow’ view defines RSS as the molecules produced from sequential one-electron oxidations of hydrogen sulfide (H_2_S), forming a thiyl (sulfhydryl) radical (HS^•^, the counterpart of HO^•^ in ROS, the same below), hydrogen persulfide (H_2_S_2_, H_2_O_2_), and the supersulfide radical (S_2_^•^, O_2_^•^) [7] (Figure 1). In the ‘broad’ view, RSS refer collectively to reactive sulfur chemotypes (both organic and inorganic) that can react with, oxidize or reduce other molecules under physiological conditions [8,9] (Figure 1). Therefore, ‘reactive’ sulfur atoms can be found in compounds having sulfur of higher valence, such as sulfite (SO_3_^2−^, +4) and sulfate (SO_4_^2−^, +6), which have interchalcogen bonds, and thiosulfate (S_2_O_3_^2−^, −1/+5 or 0/+4), in which sulfur atoms are catenated as in polysulfides (Figure 1). To date, the widely accepted notion about RSS, which we defined here as the ‘normal’ view, regards RSS as any molecules carrying reactive sulfur with a valence of −2, −1, or 0 [9,10]. In this sense, representative RSS include protein thiols (PSH) and persulfide (PSSH), low-molecule-weight (LMW) thiols (RSH) and persulfide (RSSH), hydrogen persulfide/polysulfide (H_2_S_n_, n ≥ 2), polysulfides (RSS _(n)_ R, n > 1), and sulfenic acids (RSOH) [10,11] (Figure 1). 

Formation and metabolism of RSSs have been intensively studied in eukaryotes, especially mammals [10,12]. In these processes, H_2_S is the center molecule that is generated in the cytosol through degradation of thiol-containing molecules, cysteine (cys) in particular [9] (Figure 2A). Persulfide and polysulfide can then be formed from H_2_S through various non-enzymatic reactions involving ROS. While bacteria preserve these pathways to generate H_2_S, many of them can also do so through reduction of a variety of inorganic and organic compounds containing sulfur of high higher valence, releasing H_2_S as well as other RSS as the terminal products or intermediates [13,14]. In parallel, H_2_S can be oxidized by diverse prokaryotes, and even mitochondria, through a combination of abiotic and biotic reactions to diverse sulfur compounds of any possible valence, many of which are RSS [8,9]. As inorganic sulfur in the form of sulfate (SO_4_^2−^) is extremely abundant in oceans, and metabolites in sulfur cycling are diverse, RSSs generated by microbes have profound impacts on all life and beyond [15]. 

Despite the difference in their functioning roles, the mechanism for biological activity of the reactive species is similar: modification of cellular targets. It is generally accepted that the reactive species interact with their specific cellular targets for redox signaling at physiological levels, whereas at elevated concentrations they modify macromolecules in a rather indiscriminate way, with interactions amounting to damaged enzymes and DNA, and even cell death [4]. This also holds true for RSS, although sulfur has been usually considered as the essential element constituting cellular antioxidant systems [16].

In recent years, a number of studies combining conventional genetic analysis and cutting-edge technologies have stressed the profound biological roles of RSS in mammalian as well as bacterial cells. While an understanding of the sensing, production, and physiological role of RSS in prokaryotes, bacteria in particular, is still in its infancy, it is clear that control of RSS homeostasis emerges as a promising means for therapeutic treatments for sulfur-related diseases, for maintenance of a balanced sulfur biogeochemical cycle, and for development of RSS-dependent biotechnologies. In this review, we take the notion of RSS defined by the ‘normal’ view, with an emphasis on small-molecule RSS. We discuss the origin of RSS and bacterial enzymes transforming sulfur compounds associated with RSS, as well as the functioning mechanisms of RSS and their biological impacts, and we highlight recent progress in areas of RSS sensing, which expands the understanding of RSS biology in bacteria.

## 2. Influence of Sulfur on the Origin of Life

Redox reactions, a result of electron imbalance mediated by energy, are the basis of all physiological activities in cells, such as respiration and photosynthesis. The original life began in an anoxic ferrous ocean 3.8 billion years ago (bya), likely as anoxygenic chemolithotrophs in environments associated with deep-sea hydrothermal vent systems [7,17,18]. As these environments were characterized by large amounts of sulfur and metals, iron in particular, it is believed that iron-sulfur clusters were the first electron-transfer units to be produced during chemical evolution [19]. In addition, sulfur now has been recognized to be a crucial element in the physiology of life’s common ancestor because some sulfur species, such as acetyl-coenzyme A, are essential intermediates in metabolism [20,21]. Moreover, H_2_S and diverse RSS, such as elemental sulfur, sulfite, thiosulfate, and polysulfides, rather than sulfate, which is the predominant inorganic sulfur compound on the current earth, were more prevalent in the ancient ocean and have played an critical role in shaping the sulfur cycle and life [22,23,24]. 

A large portion of contemporary living organisms, eukaryotes in particular, use O_2_ as the terminal acceptor for energy production. During oxygen respiration, ROS are generated endogenously in the cell. It has been postulated that antioxidant mechanisms, including superoxide dismutase, catalase, and peroxide reductase, evolved concomitantly with oxygen respiration [25]. However, the history of RSS is undoubtedly much longer than that of ROS, and more importantly, antioxidant mechanisms were present far earlier than the oxic ocean [26]. The anoxygenic photosynthesis appeared ~3–3.5 bya, several million years before the cyanobacterial existence, according to fossil evidence [27,28]. In the early photosynthetic organisms, their light-gathering antennae and processing machinery were not sophisticated enough to support oxidation of H_2_O, and instead, H_2_S was more likely to act as the electron donor [7,29,30]. From 2.3 billion years on, in the ‘great oxidation event (GOE)’, cyanobacteria had evolved ability to oxidize H_2_O and produced a large amount of oxygen via photosynthesis [31]. During that time, the atmospheric O_2_ fluctuated between 0%–2% for 1.7 billion years, while the oceans remained anoxic [17,29,30,32,33]. 

While the first eukaryotes appeared approximately 1.5 bya, it was not until 0.6 bya that the ocean finally became aerobic [25,34]. This time, called the ‘Boring Billion’, represents a period of geobiological stasis caused by prolonged nutrient, climate, atmospheric and tectonic stability [35]. Phylogenetic studies have shown that antioxidant enzymes were present in the last universal common ancestor, well before the emergence of cyanobacteria and the GOE [36]. These enzymes are present in all domains of life, even in anaerobic bacteria, suggesting that these enzymes are ancient and that the need for ROS protection is pervasive in non-aerobic environments [26,36,37,38,39,40,41,42]. Because of the chemical similarity between O and S, and the less electron-negativity and greater reactivity versatility of S, these enzymes have been proposed to be primarily used to resist RSS. In recent years, evidence supporting this proposal has emerged [26,43,44,45,46]. All of these insights have provided the new interpretation of RSS in biological evolution.

## 3. Transformation of Sulfur Compounds Associated with Amino Acid Metabolism in Bacteria 

Prokaryotes, especially bacteria, are renowned for metabolic diversity, which endows these organisms with unparalleled capacity in the biotransformation of sulfur compounds [47]. Through microbial metabolic processes that transform the oxidation state of sulfur, a variety of inorganic and organic sulfur compounds are generated. H_2_S, the predominant inorganic sulfur compound in earth’s earlier days that serves as the foundation for other RSS, is well-known for driving photosynthesis and energy metabolism in sulfide-oxidizing and sulfate- or sulfite-reducing microorganisms [48,49,50]. Bacterial H_2_S production has been perceived as a metabolic side product, but recent studies have demonstrated that H_2_S is an important signaling molecule and can protect bacteria from antibiotic and oxidative stress [51,52]. In bacteria, H_2_S can be generated endogenously either through amino acid metabolism or reduction of compounds having sulfur of higher valence. 

### 3.1. S^2−^ Biogenesis through Amino Acid Metabolism

There are at least three enzymes playing roles in bacterial H_2_S endogenous synthesis by utilizing homologues of the mammalian sulfur metabolic network. Two enzymes, cystathionine β-synthase (CBS) and cystathionine γ-lyase (CSE), are involved in the transsulfuration pathway [53,54,55]. The third enzyme, mercaptopyruvate sulfurtransferase (MST), converts 3-mercaptopyruvate (3-MP), derived from the reaction of cys with keto acids catalyzed by cys aminotransferase (CAT), to pyruvate and persulfide for the release of H_2_S [56] (Figure 2). 

CBS, located at the intersection of two amino acid metabolic pathways, the methionine cycle and transsulfuration pathway, is involved in the redirection of homocys from the methionine cycle toward glutathione (GSH) synthesis, finally influencing H_2_S biogenesis [53,54]. When it comes to the roles in the transsulfuration pathway, CBS catalyzes the β-replacement of L-serine or L-homocys, forming cystathionine and H_2_O [53,54,57,58]. It can also catalyze β-replacement of L-cys by H_2_O, by L-homocys, or by L-cys forming L-cystathionine, L-lanthionine, or L-serine, respectively, while releasing H_2_S in the process. As an enzyme displaying multiple functions in amino acid metabolism, CBS activity can be regulated in distinct ways, including heme-dependent allosteric regulation, adoMet-dependent allosteric regulation, and posttranslational covalent modifications [59]. Like CBS, CSE displays abundant substrate promiscuity and can catalyze the γ-elimination of cystathionine to give L-cys, α-ketobutyrate, and NH_3_·H_2_O, generating H_2_S [60] (Figure 2). In addition, CSE also produces pyruvate, Cys-SSH, homocys persulfide, etc. from L-cys and L-homocys [61,62]. Compared to CBS/CSE-dependent H_2_S synthesis, our knowledge of CAT/MST-dependent H_2_S biogenesis in bacteria is limited. It is known that transformation of 3-MP to pyruvate and H_2_S requires an intermediate of protein persulfide, E-SSH, which acts as a source of H_2_S under reducing conditions [56,63].

Generally, bacteria can encode either CBS/CSE or MST, but exceptions that carry all of these enzymes are increasingly found, such as γ-proteobacterium *Shewanella oneidensis* [64]. In addition to CBS/CSE/MST, several enzymes that contribute to H_2_S biogenesis have been identified. For example, *Fusobacterium nucleatum* has been identified to have four enzymes involved in H_2_S production, L-cys desulfhydrase, cys synthase and L-methionine γ-lyase, respectively [65,66,67,68]. In *Escherichia coli*, L-cys desulfhydrases and cys desulfurases also contribute to H_2_S production [69,70]. 

### 3.2. S^2−^ Biogenesis through Assimilatory Reduction of Inorganic Sulfur Species

In addition to generation via amino acid metabolism, H_2_S can be produced in vast amounts through reduction of compounds having sulfur of higher valence. Here we discuss the reductive pathways that release H_2_S in the cytoplasm, which can be incorporated during biosynthesis of cys. Sulfate (SO_4_^2−^), the most stable form of sulfur under current oxic conditions, is the largest sulfur pool on the Earth [71,72]. On this account, bacteria that reduce SO_4_^2−^ to HS^−^ are no doubt the most important sulfur transformers for the biogeochemical sulfur cycle (Figure 2). In parallel, L-cys synthesis from inorganic sulfate is also the major mechanism for sulfur conversion into amino acids or organic compounds, and eventually serves as a structural block for proteins [73]. The first step of sulfate reduction, which occurs in the cytoplasm and is the same for all living bacteria having this capability, converts sulfate to sulfite with adenosine-5′-phosphosulfate-reductase (APS reductase) [74]. The steps followed can be either the assimilatory sulfite reduction or the dissimilatory sulfite reduction [75]. The latter occurs in sulfate-reducing bacteria (SRB), which are composed of morphologically and ecologically diverse but physiologically unified microbes, and which will be discussed later in detail. After SO_4_^2−^ is transported into the cytoplasm, it is activated by the ATP sulfurylase to APS, which is subsequently reduced to SO_3_^2−^ by the APS reductase. The reduction of sulfite to HS^−^ through the assimilatory pathway, which was first uncovered in *E. coli* about half a decade ago, is catalyzed by the CysIJ complex [73]. The reduction costs three NADPH, generating HS^−^, which converges with O-acetyl-l-serine (OAS) on O-acetylserine sulfhydrylase A (CysK), a pyridoxal 5′-phosphate-dependent enzyme that catalyzes the final reaction of cys biosynthesis in bacteria [74,76]. 

In addition to sulfite, reduction of thiosulfate could also occur through both assimilatory and dissimilatory pathways. For assimilation, thiosulfate is up-taken into the cytoplasm of *E. coli* cells largely via the ATP-dependent transporter CysUWA, which also functions as a sulfate importer [77] (Figure 2). In addition, YeeE has recently been identified as a thiosulfate importer in *E. coli* [78]. In the cytoplasm, CysM catalyzes the reaction of thiosulfate with *O*-acetyl-L-serine (OAS) to form *S*-sulfo-L-cys, which is subsequently converted to cys upon reduction of the disulfide bond by glutaredoxin-like protein NrdH or glutaredoxin Grx1 [79,80]. Recently, a CysM-independent pathway for thiosulfate assimilation was identified, but its contribution is rather limited [81].

## 4. Dissimilatory Transformation of Inorganic Sulfur Compounds in Bacteria 

Bacteria are the predominant force driving the biogeochemical sulfur cycle, especially in oceans [47]. Through microbial metabolic processes that transform the oxidation state of sulfur, a variety of inorganic and organic sulfur compounds are generated. Although organic sulfur has profound impacts on living organisms and ecology, it plays only a minor role in the biogeochemical sulfur cycle [74,82]. As a large portion of intermediates formed in the sulfur cycle are RSS, it is conceivable that they are collectively indispensable for the biogeochemical sulfur cycle. In addition, many bacteria can harness energy released from the redox reactions constituting the biogeochemical sulfur cycle to support growth in environments. 

### 4.1. Reduction of S^6+^(SO_4_^2−^)/S^4+^(SO_3_^2−^)

In SRB, after the reduction of SO_4_^2−^ to SO_3_^2−^ completes in the cytoplasm, the dissimilatory sulfite reductase (Dsr) catalyzes the subsequent reaction that generates S_2_O_3_^2−^, and releases HS^−^ into the surroundings [75] (Figure 2). Intriguingly, although SRB are ubiquitous and Dsr has been used as the functional marker for many years, the mechanistic enigma of Dsr enzymes has been unveiled only recently [72,83,84]. For many years, Dsr enzymes have been believed to be composed of two proteins, DsrA and DsrB, in the form of hetero-tetramer [85]. Each A/B heterodimer harbors 2 siro-hemes and 2 [4Fe–4S] clusters that are presumably involved in electron transfer to HSO_3_^−^ (Figure 3A). Now it is clear that DsrA and DsrB are the minimum requirement for sulfite reduction, and additional and liable subunits, DsrC in particular, are also involved in the process [85]. The current understanding of the mechanism is that the DsrAB complex catalyzes the reduction reaction, resulting in the production of S_2_O_3_^2−^ and formation of a persulfide bond with a cys residue on DsrC [85]. Subsequently, through a yet-unknown mechanism, DsrC expunges the bound sulfur as HS^−^ and becomes oxidized, which subsequently returns back to the reduced form by accepting electrons from the cytoplasmic membrane for the next reaction [76]. In this way, DsrC couples the reduction reaction with energy conservation (Figure 2). This model also suggests more accessary proteins to be implicated in the sulfite reduction, at least a quinone dehydrogenase or equivalent at the cytoplasmic membrane that reduces the oxidized DsrC. It has been proposed that the multiple-component DsrMKJOP complex likely plays this role [72,74,86,87] (Figure 2). 

Besides SRB, various bacteria unable to respire sulfate can still use extracellular sulfite as an electron acceptor to support growth. To date, two types of non-Dsr dissimilatory sulfite reductases that are not homologous to DsrAB have been identified and characterized, differing from each other in cofactors used for catalysis [72]. While the reductases from *Salmonella enterica* and *Methanocaldococcus jannaschii* are siroheme-dependent in the manner of DsrAB, species in *Campylobacter*, *Sulfurospirillum*, *Wolinella*, and *Shewanella* have evolved cytochrome (cyt) *c*-based sulfite-reducing enzyme [88,89].

### 4.2. Reduction of S^2+^(S_4_O_6_^2−^) 

It has been known for nearly a century that some bacteria, such as *S. enterica*, could grow by respiring tetrathionate as an electron acceptor [92,93,94] (Figure 4). The product of 2-electron reduction of tetrathionate is thiosulfate [72]. To date, some distinct types of tetrathionate reductases have been described, including Ttr of *S. enterica* and TsdA of *C. jejuni* [72,93] (Figure 3B,C). Ttr is composed of three subunits: TtrA contains a molybdopterin guanine dinucleotide cofactor and a [4Fe–4S] cluster, TtrB is associated with four [4Fe–4S] clusters, and TtrC is an integral membrane protein functioning to oxidize quinol (Figure 4). In addition, TsdA has been identified as a bidirectional enzyme that catalyzes the interconversion of tetrathionate and thiosulfate [14,95,96,97] (Figure 3B,C). For oxidation of thiosulfate, the enzyme is called thiosulfate dehydrogenase (Tsd), which will be discussed later in thiosulfate oxidation. 

Although tetrathionate reduction contributes to the sulfur cycle less significantly than does sulfate/sulfite reduction, it is prevalent in the gut and is important for *S. enterica* during infection [94]. In the mammalian host, tetrathionate is formed in a large amount from oxidation of thiosulfate by ROS, which is generated during the oxidative burst upon intestinal inflammation. The resulting tetrathionate provides *S. enterica* with a selective growth advantage in the gut over the commensal microbiota lacking this capacity [94]. Interestingly, it has also been reported that the application of a relatively simple tetrathionate salt-based molecule exerts a protective effect against ischemia-reperfusion ROS-derived injuries in mammals [98].

### 4.3. Reduction of S^0^

Reduction of sulfur to HS^−^ is carried out by sulfur reductase in a respiratory type of metabolism [99]. As zero-valent sulfur (ZVS), in addition to elemental sulfur, can be found in various sulfur compounds, such as thiosulfate and polysulfides, substrates of sulfur reductases are diverse. Notably, there are two definitions for valence of two sulfur atoms of thiosulfate: the broadly accepted version of 0/+4, and as appearing in some publications as −1/+5 [8,100].

Dissimilatory sulfur reductases are widespread in prokaryotes, but they are distinct between bacteria and archaea [99,101,102] (Figure 4). Bacterial sulfur/thiosulfate reductases, such as PhsABC of *S. enterica* and PsrABC of *Wolinella succinogenes*, are composed of three subunits, a catalytic unit (A), an iron–sulfur protein (B), and an integral membrane protein (C) that serves to anchor the other subunits on the membrane [103,104]. Despite this, these enzymes may not be identical, as PhsABC catalyzes reduction of elemental sulfur and the zero-valent sulfur of thiosulfate to S^2−^, while PsrABC is capable of reducing polysulfides (S_n_^2−^, n > 2) to S_n-1_^2−^ and HS^−^ in a stepwise mode [103,105]. 

### 4.4. Oxidation of S^2−^ and S^0^


Sulfide generated from sulfate reduction can be converted to more oxidized sulfur species, either biotically or abiotically, including sulfur, thiosulfate, sulfite, and sulfate, with sulfate making up the vast majority (>90%) [72,106] (Figure 2 and Figure 4). Like reduction, oxidation of sulfide in living cells can be coupled to energy conservation by supplying electrons to the respiratory electron transport chain [107]. Oxidation of sulfide to sulfane sulfur is the most important process for formation of RSS, such as hydrogen persulfide/polysulfide and organic persulfide/polysulfide. Sulfane sulfur is a common cellular component, maintained in a micromolar range, and changing with growth stages in bacteria [108,109]. Most cellular persulfides and polysulfides are generated from sulfide (S^2−^) oxidation by SQRs and flavocyt *c* sulfide dehydrogenases (FCSDs) [110,111], or LMW persulfides from the metabolism of L-cys by MST and cysteinyl-tRNA synthetase [112,113]. H_2_S oxidation can be catalyzed by the concerted actions of three enzymes: SQR, PDO, and Rhd [114,115]. The work model established with Gram-negative bacterial SQR systems suggests a series of sequential reactions, including oxidation of sulfide to polysulfide by membrane-bound SQR, formation of GSSH from spontaneous reaction between polysulfide and GSH, oxidation of the sulfane sulfur in GSSH to sulfite catalyzed by PDO, and formation of thiosulfate from spontaneous reaction between polysulfide and sulfite [116,117] (Figure 4). Rhd, a thiosulfate sulfurtransferase that tends to speed up the reaction of polysulfide with GSH to produce GSSH, is not essential, as the reaction can occur spontaneously [104,107]. Although wildly distributed, SQRs have been proposed as primarily serving as sulfide detoxifiers, as they are present in mitochondria of many eukaryotes [70] (Figure 2). While SORs are structurally diverse, they can be conveniently grouped into two types according to their PDO partners, a well-known member of 2His–1Asp mononuclear iron-containing enzyme superfamily [110,118,119,120]. Type I includes PDOs in mitochondria and heterotrophic bacteria, and therefore, SQRs working with this type of PDOs function in the cytoplasm (Figure 2). Type II is found in dissimilatory sulfur-oxidizing bacteria, such as *Rhodobacter capsulatus*. In this case, PDOs are located outside the cytoplasmic membrane, and, consistently, SQRs catalyze the reaction at the outer surface of the cytoplasmic membrane [121,122,123] (Figure 4). In some bacteria, PDO and Rhd are naturally fused into a single polypeptide, becoming a bifunctional enzyme [124,125]. In addition to PDO and Rhd, CstB also plays a role in sulfane sulfur (RSSH) oxidation, converting two equivalents of persulfide to thiosulfate as a final product [124]. 

Flavocyt *c* sulfide dehydrogenases (FCSDs) represent another group of sulfide oxidizing enzymes in bacteria [111,114,126] (Figure 4). FCSDs, widely distributed in diverse bacteria, comprise two subunits in the periplasm, a large sulfide-binding flavoprotein and a small cyt *c* [114]. The final production of sulfide oxidation catalyzed by FCSDs is polysulfide [111]. It should be noted that many bacteria are equipped with multiple sulfide-oxidizing systems, including both SQR and FCSD [127]. Although SQR is generally regarded as the predominant sulfide-oxidizing system, FCSD may confer cells carrying both systems advantages under certain growth conditions. 

### 4.5. Oxidation of S^0^ and S^2+^

In sulfur-oxidizing (Sox) prokaryotes, the best-studied system for sulfur oxidation is the Sox multienzyme system located in the periplasm [72] (Figure 4). The Sox systems may have promiscuity for substrates as they have been reported to carry out oxidation of sulfide and sulfite in certain bacteria [114,128]. Nevertheless, the primary role that the Sox system plays is to catalyze oxidation of zero-valent sulfur (ZVS) from a variety of sulfur compounds [72]. Among them, thiosulfate is considered particularly important, as it is a main product of most S^2−^/S^0^ oxidation and fulfills a key role in the sulfur cycle [14]. In *Paracoccus pantrophus*, whose Sox system has been intensively studied, the *sox* gene cluster comprises at least two transcriptional units with fifteen genes, seven of which, *soxXYZABCD*, encode proteins essential for sulfur oxidation in vitro [49,129,130]. The Sox pathway initiates with the activation of unconjugated SoxYZ with thiosulfate involving both SoxAX and SoxB, resulting in a SoxYZ-S-sulfane adduct via intermediate SoxYZ-S-thiosulfonate [131,132,133] (Figure 4). Once activated, the oxidation cycle commences. A disulfide bond between the sulfane sulfur of thiosulfate and the persulfurated active site cys residue on the carrier arm of SoxYZ is formed by SoxAX, where heme 2 is an active site (Figure 3D left), generating SoxYZ-S-thioperoxosulfonate, which subsequently is converted to SoxYZ-S-thiosulfane after releasing a molecule of sulfate under the catalysis of SoxB. For catalysis, the carrier arm of SoxY has to be positioned at the substrate channel of SoxB, leading to formation of a disulfide bond between Cys151 of SoxY and Trp175 of SoxB (Figure 3D, right). This reaction catalyzed by SoxB also generates the SoxYZ-S-sulfane, which acts as the activated adduct to start a new round of thiosulfate oxidation [131,132]. 

In many organisms, this aerobic Sox pathway lacks certain components, depending on species, and can only oxidize sulfide to soluble elemental sulfur [49,134]. For example, *Rhodobacter* spp. converts sulfide to sulfur using a Sox system without SoxCD [114]. The resulting zero-valent sulfur can be either released to the surroundings or stored in the periplasm as sulfur globules, a subject which has recently been reviewed in detail by Dahl [72]. Although the chemical nature of the sulfur in the globules is still under debate, it is increasingly accepted that octasulfur (S_8_) in a nano-crystalline form may be the main representative [72,135]. In addition, the truncated Sox system may work in conjunction with other sulfur oxidation pathways to complete oxidation of thiosulfate to sulfate [70]. Among them, the cytoplasmic sulfur oxidation pathway involving TusA is the first to be identified. The TusA protein, which has been identified as a central element supplying and transferring sulfur as persulfide to a number of important biosynthetic pathways, is implicated in providing elemental sulfur imported from the extracellular space to DsrABC for oxidation to sulfite [119,136,137]. However, despite recent advances, the mechanisms underpinning the transport of sulfur into the cytoplasm and the TusA involvement have to await further exploration. Recently, two other pathways in sulfur-oxidizing prokaryotes have been identified to complement the truncated Sox system for oxidation of thiosulfate to sulfite. One is composed of the proteins resembling the HdrA, B, and C subunits of heterodisulfide reductase from methanogenic archaea [138,139]. This pathway may require TusA-like proteins as well, and, more importantly, a lipoate-binding protein [140]. Given the indispensability of the lipoate-binding protein, it has been suggested that reduction of lipoate to dihydrolipoate is a part of the reaction cycle [140]. The other is a mixture of SoxB, TsdA and cytoplasmic sulfur dioxygenase (SDOs) [100]. In this pathway, TsdA converts thiosulfate to tetrathionate, from which sulfone is released by SoxB to form zero-valent sulfur, and subsequently zero-valent sulfur is oxidized to sulfite by SDOs. It is worth noting that in many sulfur oxidizers, SDOs function independently to oxidize sulfane sulfur bound to GSH (GSSH, and GSS_n_H) [141,142]. 

Thiosulfate can also be oxidized to tetrathionate, a process identified in diverse prokaryotes [72] (Figure 4). Enzymatic systems capable of catalyzing this reaction include thiosulfate dehydrogenase (Tsd) and partial Sox complex, representing a good interpretation of the branched thiosulfate oxidation [143]. Tsd, a widespread and well-studied system, is commonly composed of two functional diheme cyt *c* subunits TsdA and TsdB, which may be fused into a single polypeptide in some bacteria, such as *M. purpuratum* [96,144]. TsdA and TsdB function as the catalytic subunit and the electron acceptor partner respectively [145]. As mentioned earlier, TsdA is in fact a bidirectional enzyme that also acts as a tetrathionate reductase in some bacteria, such as *C. jejuni*, in which TsdB is missing [95,97]. The crystal structure of TsdA reveals His/Cys iron coordination for Heme 1, the active site of the enzyme, to which a thiosulfate covalently can be bound via the cys [96,145,146]. The diheme cyt *c* TsdB, which may be fused to TsdA, is the electron-accepting unit for transport electrons abstracted from the oxidation to cyt *c* heme-copper oxidase (HCO) of the respiratory chain for energy generation, although the exception has been found [14,96] (Figure 3B,C). Comparison of the structure of *C. jejuni* TsdA predicted by AlphaFold to a structurally available counterpart catalyzing thiosulfate oxidation reveals a high level of structural similarity. However, *C. jejuni* TsdA has a long random coil in the N-terminus (Figure 3C). According to phylogenetic and genomics analyses, the enzymes identified as thiosulfate dehydrogenase are distributed more broadly [14,96,145] (Figure 4).Intriguingly, some bacteria, such as *S. oneidensis*, are equipped with independent enzymatic systems catalyzing oxidation and reduction of thiosulfate [14]. 

In addition, thiosulfate oxidation can occur as an intermediate step in the oxidation of reduced sulfur compounds to sulfate, called tetrathionate (S_4_O_6_^2−^) intermediate (S_4_I) pathway [134,147,148]. In the S_4_I pathway, the conversion of thiosulfate to tetrathionate is catalyzed by thiosulfate:quinone oxidoreductase TQO [149]. Unlike Tsd, the TQO enzyme transfers electrons into the respiratory chain via quinones, and beyond this, little is known about the catalytic mechanism of the enzyme. 

In prokaryotes, oxidation of tetrathionate to sulfate, whether it is a part of the S_4_I pathway or not, has been known for some time [148,149,150,151,152]. Tetrathionate hydrolase, TetH, is responsible for hydrolysis of tetrathionate to sulfate and disulfane monosulfonic acid (−S-S-SO_3_^−^) in the S_4_I pathway [122,150,153]. Disulfane monosulfonic acid is a highly reactive sulfur species, which either quickly decomposes to sulfur and thiosulfate or autocombines into long-chain sulfur compounds, ultimately resulting in the production of elemental sulfur and sulfite. Another enzyme known to oxidize tetrathionate is thiol dehydrotransferase from *Advenella kashmirensis*, which is independent of S_4_I [152]. This enzyme is able to catalyze two sequential reactions, the formation of tetrathionate from thiosulfate and the reaction of tetrathionate with reduced GSH. The resulting product, GSH sulfodisulfane adduct, can be fed into the thiosulfate oxidation cycle catalyzed by Sox [152].

### 4.6. Oxidation of S^4+^

Sulfite is a highly reactive nucleophile and could cause damage to DNA and proteins through reacting with disulfide bonds [154,155]. Although sulfite can be spontaneously oxidized to sulfate in oxic environments, in order to protect cells against sulfite-induced damage, virtually all forms of living organisms are capable of performing biotic oxidation of sulfite [155,156]. In addition, sulfite has recently been reported to be protective against oxidative stress induced by high concentrations of glutamate called oxytosis in mammals [157], which has recently been suggested to be equivalent to ferroptosis [158]. In bacteria, the prevailing route for sulfite oxidation, called the direct route, is carried out by molybdenum (Mo)-containing sulfite-oxidizing enzymes (SOEs) in the periplasm [159] (Figure 4). All of the SOEs possess a *cis*-dioxo MoO_2_^2+^ center that is bound to a unique pterin, molybdopterin, which is essential for catalysis [160,161]. Bacterial SOE, commonly termed as sulfite dehydrogenase (SDH), differs from its eukaryotic counterpart, which is a single polypeptide enzyme, in that it requires a cyt *c* subunit as electron acceptor in addition to the Mo-containing catalytic subunit [156,162]. The electrons extracted from the sulfite oxidation process are transferred to the HCO oxidase of the respiratory chain via the cyt *c* subunit for energy generation [163,164]. 

Oxidation of sulfite to sulfate can also occur in the bacterial cytoplasm (Figure 2). The enzymatic complex, SoeABC, first identified in *Allochromatium vinosum* but universally present in bacteria, comprises cytoplasmic molybdoprotein SoeA and iron-sulfur protein SoeB, as well as membrane-bound SoeC [127,147]. In many bacteria, SoeABC appears to be the major sulfite-oxidizing agent. Less commonly, the oxidation of sulfite to sulfate can be achieved via reversing the sulfate reduction pathway [72,155]. 

## 5. Physiological Impacts of RSS

Investigations into the physiological roles of RSS in bacteria are primarily centered on H_2_S, largely due to its physiological and pharmacological effects as a redox signaling molecule in mammals [9]. Despite the beneficial role of H_2_S at low concentrations, at high concentrations (millimolar), it is toxic to virtually all living organisms [9]. H_2_S is a highly poisonous gas, as a cause of inhalational deaths second only to carbon monoxide (CO), and this reputation has been known for centuries [165]. All other RSSs exhibit similar concentration-dependent effects on the physiology of living organisms. Not surprisingly, elemental sulfur and inorganic polysulfides have been explored as a weaponry to inhibit various types of pathogenic and drug-resistant microorganisms [44,166]. 

Persulfidation of proteins by H_2_S has been repeatedly reported as an important means in signaling, but this process is still under debate because H_2_S *per se* cannot react directly with thiols or does so after autooxidation in a sufficiently rapid manner [8]. Nonetheless, it is clear that disulfide bonds within proteins can be easily subjected to persulfide modification through a nucleophilic attach by the sulfide anion [8]. This may be particularly significant in prokaryotes because such proteins are abundant in the oxidizing milieu of the periplasm where the sulfide anion generated biotically and abiotically is present [8]. Nonetheless, some other RSSs, especially sulfane sulfur, including those in persulfide and polysulfide forms, and elemental sulfur (S_8_), which are more active than H_2_S, have been shown to be the leading agents for protein persulfidation [167,168,169]. 

Previous work in mammals has demonstrated that protein persulfidation affects specific activity in either positive (activation) or negative (inactivation) way, modulating diverse biological processes [170,171,172]. Upon sulfane sulfur treatments, persulfidation of the *Staphylococcus aureus* proteome is widespread [173]. The treatments increase the number of the modified proteins by over 30%, from 238 to 305. In addition to proteins involved in transcriptional regulation and metabolism (will be discussed later in the section of RSS sensing), the study reveals many secreted virulence factors and two uncharacterized thioredoxin-like proteins. 

### 5.1. Inhibition of Energy Conservation and Growth

The ultimate phenotype of H_2_S at high concentrations is growth inhibition, which has been seen from a variety of bacteria, such as *E. coli*, *S. oneidensis*, *Acinetobacter baumanni*, to name a few [64,174]. In humans, the primary target of H_2_S is the mitochondrial oxygen reductase, a cyt *c*-type HCO which carries out reduction of O_2_ to H_2_O using electrons from soluble cyt *c* [174,175]. HCOs are also commonly found in bacteria, but in different forms, including cyt *aa*_3_ oxidases (mitochondrial-like oxidases) as in *Bacillus subtilis* and *S. aureus*, cyt *bo*_3_ oxidases as in *E. coli*, and cyt *cbb*_3_ as in *S. oneidensis* and *Pseudomonas aeruginosa* [100]. While all of HCOs employ the proton-pumping mechanism during oxygen reduction to generate energy efficiently, they are highly sensitive to a variety of small molecules, including H_2_S, CO, HCN, and nitrite/nitric oxide (NO) [100,176,177,178,179].

H_2_S inhibits HCOs in a biphasic manner, non-competitive with respect to either O_2_ or cyt *c* (for cyt *c*-type HCOs) [180]. The first molecule of H_2_S (in the form of HS^−^) binds to Cu_B_ in either cupric or cuprous state, and then is transferred to the ferric heme *a_3_*, resulting in inactivation of the catalytic activity of the enzyme [180,181]. Subsequently, the second molecule of HS^−^ interacts with the enzyme to form the final protein-inhibitor adduct, in which heme *a* and Cu_A_ likely stay reduced [180]. During the reaction, H_2_S is probably oxidized to persulfide species. Unlike CO or NO, H_2_S may not react with fully reduced enzyme, and thus HCOs in the oxidized or mixed valence states are probably involved in both inhibitory and metabolic reactions with sulfide [180].

Similarly, H_2_S rapidly and effectively inhibits the activity of quinol-type HCOs, such as *E. coli bo*_3_ [182]. The apparent half-maximal inhibitory concentrations of H_2_S for cyt *c*-type and quinol-type HCOs are comparable, 0.55 and 1.1 μM, respectively [182,183]. Interestingly, the inhibition of the *bo*_3_ oxidase is fully reversible [182]. Although such an observation was not reported before with cyt *c*-type HCOs, it is probably common to all HCOs. 

The inhibitory effect of H_2_S on bacterial growth is well recognized, but the molecule usually does not kill bacterial cells, contrasting with its lethality for mammals. One explanation would be that prokaryotes commonly carry additional terminal oxidases, the *bd*-type quinol oxidases in particular [179]. The *bd* oxidase lacks a counterpart proton-pumping mechanism as in HCOs and is not efficient in generation of proton motive force [184,185]. Consequently, unlike HCO, whose main role is to conserve energy, the *bd* oxidase endow bacterial cells with resistance to various harmful molecules, including H_2_S [179,186,187,188]. It has been found that the *bd* oxidases of *E. coli* remain active in the presence of H_2_S at 58 μM, indicating that the *bd* oxidases are insensitive to H_2_S [182,188]. Moreover, the difference of the oxidases in resistance to H_2_S may have profound impacts in bacterial physiology. It has been found recently that endogenous H_2_S alters energy metabolism and growth of *Mycobacterium* species by modulating activities of different oxidases [189,190].

### 5.2. Role of H_2_S in Oxidative Stress Response and Immune Response

H_2_S alone exerts little bactericidal activity against bacterial cells, but when combined with hydrogen peroxide (H_2_O_2_), it mediates a dramatic increase in cytotoxicity [64] (Figure 5A). This observation suggests that the cellular targets subjected to inhibition of H_2_S are beyond HCOs in bacteria. One of the key targets is catalases, most of which are hemoproteins primarily responsible for scavenging H_2_O_2_ [191,192]. It is well established that H_2_S can interact with several metals, including iron, copper, nickel, and zinc [193]. In the presence of H_2_O_2_, the interaction of H_2_S with the heme iron complex of hemoproteins leads to formation of sulfheme (sulfur-incorporated porphyrin) iron complex, resulting in loss of enzymatic activity [194,195]. Without protection of catalases, the bacterial cells are quickly killed by H_2_O_2_ [64,196,197] (Figure 5A). It is worth mentioning that an additional mechanism underlying the inhibition of catalases by H_2_S has been identified in plants, which is protein persulfidation, a result of the interaction between H_2_S and thiol groups of targets [198]. 

Interestingly, the combinatorial redox action of H_2_S and H_2_O_2_ that promotes cytotoxicity is just one side of the story. A contrasting effect has been seen, even if the timing of application is different (Figure 5A). When applied sequentially, H_2_S, like NO, acts as a molecule protecting *E. coli*, *Staphylococcus*, and *Bacillus* from ROS-mediated killing [51,199,200]. A similar phenomenon has also been observed from the combination of NO and H_2_O_2_ against a variety of bacteria [51,64]. Under this circumstance, bacterial cells promptly activate OxyR (or its equivalent), the master transcriptional regulating modulating cellular oxidative stress response. As a result, production of all sorts of ROS scavengers and damage-control proteins is drastically enhanced, leading to an increase in the bacterial tolerance to ROS [64,201] (Figure 5A). Given that oxidative stress is intertwined with antibiotic susceptibility, endogenous H_2_S mediates the sensitivity of various bacteria to a range of antibiotic agents [51]

Most recently, H_2_S has been recognized as playing a crucial role in human health by regulating immune response against bacterial pathogens. On one hand, endogenous H_2_S produced by cells of *E. coli* and *S. aureus* at elevated levels increases the resistance to immune-mediated killing [202]. Consistently, inhibitors of bacterial H_2_S biogenesis potentiate bactericidal antibiotics against *P. aeruginosa* and *S. aureus* in vitro and in mouse models of infection [203]. On the other hand, H_2_S released by host cells also affects pathogenesis [204]. In mice lacking CSE, cells of bacterial pathogen *Mycobacterium tuberculosis* survive less and have reduced colony forming units. This compromised pathogenesis by host H_2_S is a result of altered central metabolism, including glycolysis and the pentose phosphate pathway [204]. While many questions remain unanswered, all of these new findings manifest that H_2_S homeostasis during pathogenesis is critically linked to immunometabolism. 

The physiological overlap and/or cross-talk between RSS and RNS, especially H_2_S and nitric oxide (NO), has come to the frontline of research in recent years and is reviewed in detail by Ivanovic-Burmazovic and Filipovic [205]. This is conceivable, as all of these compounds impact protein activity through thiol modifications, either synergistically or antagonistically [206]. In accord with this, thionitrous acid (HSNO), the smallest *S*-nitrosothiol formed from reaction of sulfide with low-molecular weight and/or protein S-nitrosothiols, including NO, is found to be the key molecule in cell signaling [207,208,209]. In mammals, it has been reported that the interaction of H_2_S and NO produces polysulfides (H_2_S_n_), which serves as an activating molecule for TRPA1 channels [210]. In addition, organic thiols (RSH) can react with nitrite (NO_2_^–^) to form organic nitrosothiols (RSNO), which readily react with H_2_S at acidic pH to form a mixture of polysulfides [211]. Physiologically, nitroxyl donors via formation of HSNO and its cousins, such as nitrosopersulfide (SSNO^–^), could elevate cellular levels of RSS in *S. aureus* [212,213]. Despite these insights, impacts of the interplay of RSS and RNS on the biology of prokaryotes remain largely unknown. 

### 5.3. Role in Metal Reduction 

Electroactive microorganisms known to date, mostly bacteria, are renowned for their respiratory versatility, are capable of respiring an array of diverse chemicals as electron acceptors, and play a critical role in the biogeochemical cycle of elements, including metals, such as iron [214,215,216]. As a large portion of minerals are insoluble, these microorganisms have evolved multiple strategies for extracellular electron transfer, through which the electrons abstracted from oxidation of electron donors are transferred to multivalent metal ions within minerals [214]. Many bacteria, among which *S. oneidensis* is best studied, use porin-cyt *c* complexes to transport electrons from the cytoplasmic membrane to the outer cell surface. Nanowires, electrically conductive pili in which cyt *c* proteins are stacked, as best illustrated in *Geobacter sulfurreducens*, represent another strategy; while this electron transport machinery is found less commonly, it could deliver electrons to minerals at a distance. Additionally, soluble electron shuttles can be utilized alone or in conjunction with the first two means [147,215,217,218]. 

Despite these effective strategies, metal reduction in nature by electroactive bacteria could be drastically affected by environmental parameters. These microorganisms, including research models *S. oneidensis* and *G. sulfurreducens*, usually have exceptional ability to transform sulfur species. For example, *S. oneidensis* is able to not only reduce sulfite, thiosulfate, tetrathionate, and elemental sulfur to H_2_S, but also to oxidize thiosulfate to tetrathionate [14,64,89,108,219]. Sulfur species, generated from sulfur recycling in natural systems through a number of oxidation, reduction and disproportionation reactions both biotically and abiotically, have been demonstrated to be indispensable players in metal reduction [220,221,222,223]. Center to this sulfur-mediated metal reduction is HS^−^, which is produced by microbial reduction of various sulfur species and reacts with Fe^3+^ to form FeS (Figure 5B). This abiotic reaction between HS^−^ and Fe^3+^ not only accounts for reduction of a large quantity of iron oxides but also offers an explanation of the low concentrations of HS^−^ in aquifers, in addition to the oxidation to sulfate [220,224]. Given that these intertwined reactions of the sulfur and iron cycle are in part biological, electroactive microorganisms that maintain both iron and sulfur metabolic pathways may have an impact in ecology more substantially than previously expected. 

## 6. RSS Sensing and Regulation

In order to survive and thrive in their natural niches, bacteria have evolved diverse adaptive strategies to swiftly respond to various adverse stress conditions, and transcriptional regulation is key for this adaptation [225]. As signal molecules, RSS tend to trigger modification of specific targets via protein persulfidation and subsequently affect the down-stream biological processes. To combat RSS at high concentrations, cells have various enzymes and regulatory systems to protect against excessive levels of modification [43]. A common set of proteins capable of cleaning or reducing the modification include thioredoxin and/or peroxiredoxin, and this scenario appears to be conserved in bacteria [46,226,227]. In *Synechococcus* sp. PCC7002, all of six peroxiredoxins are induced by S_8,_ and peroxiredoxin I (PrxI) has been experimentally demonstrated to be able to reduce S_8_ to H_2_S [46]. To regulate the expression of the genes encoding common sulfide detoxification or repairing enzymes, a variety of mechanisms governing cellular responses to RSS have been identified in bacteria, among which transcriptional regulation by DNA-binding proteins is the most prevalent and important. 

In *S. aureus*, *cstR* (CsoR-like sulfurtransferase repressor), located in *cts* operon, is the first gene reported to be involved in persulfide-responsive transcriptional regulation, which modulates the expression of five other *cts* genes to function [228,229]. CstR is an all-helical protein characterized by a disc-shaped D2-symmetric or pseudosymmetric tetrameric architecture and homotetrameric in solution, harboring four peripheral dithiol sensing sites (Figure 6, left). This regulator responds to cellular sulfide stress through sensing LMW persulfides and inorganic polysulfides (Sn^2–^), forming a mixture of di-, tri-, and tetra-sulfide interprotomer linkages between Cys^31^ and Cys^60^ [59,213,228]. Despite this, the information about DNA recognition and the changes upon inducer binding in CstR, along with other members of the CsoR family, which was defined recently, is rather limited [230]. CstR in bacteria, especially in the intestinal ones, such as *Enterococcus faecalis*, has been proved important to sustain sulfur-homeostasis as the intestinal flora is the major producer of L-homocys, one of the constituents of the LMW thiol pool [213,231,232].

Another sulfide-related regulator identified to date is SqrR (sulfide:quinone redcutase repressor) of *R. capsulatus*, a member of the ArsR family of bacterial repressors, which are widely distributed in Proteobacteria [225,233]. Experiments in vitro showed SqrR forms a dimer (Figure 6 right) and senses sulfide via an intramolecular tetrasulfide-bond formed between two conserved cys residues, Cys^41^ and Cys^107^ [234,235]. A homologue of SqrR, BigR (Biofilm growth-associated repressor), characterized in *Xylella fastidiosa* [236,237,238] and *Acinetobacter baumannii* [227], regulates a secondary RSS detoxification system [227,239]. BigR is responsive to H_2_S via forming a disulfide bond between Cys^42^ and Cys^108^ [234]. PigS, another member of the ArsR family [239], regulates part of the PigP regulon involved in the biosynthesis of the antibiotic prodigiosin (Pig), implying a regulatory connection between antibiotic biosynthesis and H_2_S/RSS homeostasis in *Serratia* [59]. The *pigS* gene *per se* is directly under transcriptional regulation of PigP, a novel master regulator that is implicated in secondary metabolism in *Serratia* and whose homologues are found only in a very limited group of Enterobacteriaceae [59]. Additionally, FisR (Fis family transcriptional regulator) acts as a RSS-responsive activator in *Cupriavidus pinatubonensis*, sensing RSS via forming disulfide and tetrasulfide cross-links among three conserved cys residues [235,240]. 

To date, a large number of studies have demonstrated physiological cross-talks between ROS and H_2_S/RSS [59,173,241,242]. Conceivably, the systems responding to ROS might also be able to sense RSS. OxyR, the master transcriptional regulator mediating cellular responses to H_2_O_2_-induced oxidative stresses in a variety of bacteria, has been found to be able to sense sulfane sulfur [43]. Most recent investigations have revealed that LasR and OhrR of *P. aeruginosa*, the primary transcriptional regulators in quorum sensing and the response to organic peroxides respectively, can also sense sulfane sulfur [243,244]. Thus, RSSs, especially sulfane sulfur, represent a group of new signaling factors that are involved in diverse biological processes, whose regulation is dependent on both RSS-specific regulators and global regulators primarily responsive to other environmental cues.

## 7. Concluding Remarks

In this review, we have summarized recent advances in generation of H_2_S and its derivative RSS via amino acid metabolism and biological transformation of inorganic sulfur species, their influence on the physiology, and how they are perceived in bacteria. In both higher organisms and prokaryotes, the amino acid metabolism as the source of H_2_S has been regarded to be the most critical one for the beneficial roles of H_2_S, signaling in particular. On the contrary, avast amount of H_2_S and RSS can be produced through biological transformation of inorganic sulfur compounds, which not only predominantly drives the global sulfur biogeochemical cycle, but also imposes a threat to cells. It is clear that the processes with sulfur compounds as electron donors and/or acceptors are early contributors to energetics prior to O_2_. To date, we have known a great deal about the enzymes that catalyze most, if not all, of steps of the transformation. Nevertheless, new enzymes and novel functioning mechanisms of known enzymes are continuingly revealed, especially in recent years. 

The inherent chemical reactivity of H_2_S and RSS, especially when in conjunction with other reactive species, challenges development of all lifeforms. Redox-sensitive proteins are primary targets of H_2_S and RSS because of their susceptibility to modification by the formation of protein-inhibitor adducts. Consequently, the redox status of the cell, and the factors accounting for maintaining redox homeostasis, including those involved in combating ROS and RNS stresses, are particularly important in promoting or antagonizing the effects of H_2_S and RSS. In recent years, multiple lines of evidence have substantiated that bacterial cells have developed a sophisticated system to maintain homeostasis of H_2_S and RSS, many members of which are transcriptionally responsive to H_2_S and RSS. Furthermore, recent advances also convince us to consider H_2_S and RSS as a promising drug in promoting the effectiveness of antimicrobials and beyond. Nonetheless, the field of RSS chemical biology is still in its infancy, with many more questions open than answered. This may be particularly true with prokaryotes, as they are so diverse in phylogeny and metabolism. For instance, cellular targets of RSS are likely different in bacteria, even at the level of species and strain. Thus, with the availability of enormous amounts of genomic sequences and data in various omics, bacteria are no doubt a repertoire of living organisms for unraveling new mechanisms of RSS formation and metabolism impacting cellular signaling and function.

## Figures and Tables

**Figure 1 antioxidants-11-02487-f001:**
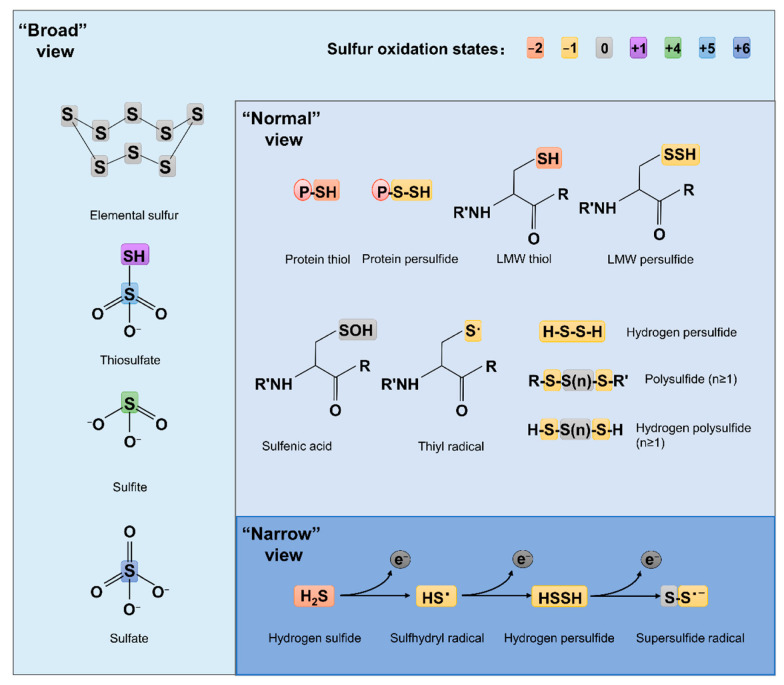
Structures of some RSS chemotypes in different views. The red (−2), yellow (−1), gray (0), purple (+1), green (+4), light blue (+5) and dark blue (+6) rectangles are used to designate the valence states of sulfur, as specified.

**Figure 2 antioxidants-11-02487-f002:**
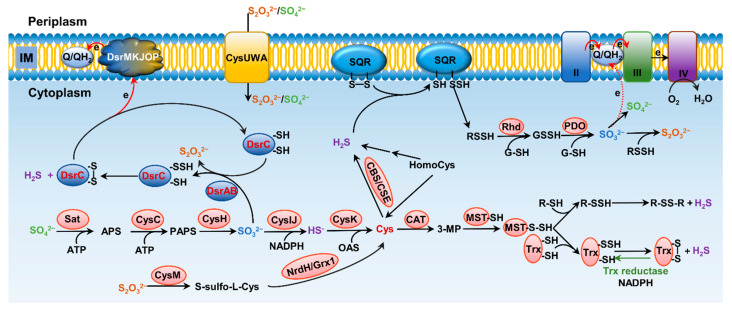
Pathways for bacterial sulfur metabolism in the cytoplasm. H_2_S biogenesis through amino acid metabolism: generation of H_2_S from homocys can be catalyzed by cystathionine β-synthase (CBS) and cystathionine γ-lyase (CSE). Cys aminotransferase (CAT) catalyzes the formation of 3-mercaptopyruvate (3-MP) from cys, and then, mercaptopyruvate sulfurtransferase (MST) converts 3-MP to H_2_S. H_2_S biogenesis occurs through assimilatory sulfate reduction (ASR) and dissimilatory sulfate reduction (DSR) of inorganic sulfur species; the latter only occurs in sulfate-reducing bacteria (SRB), catalyzing SO_3_^2−^ to HS^−^ through dissimilatory sulfite reductase (Dsr). For ASR pathway, SO_4_^2−^, which imported from ATP-dependent transporter CysUWA, is catalyzed and converted to HS^−^ by a series of enzymes, including Sat, CysC, CysH, and CysIJ. S_2_O_3_^2−^ can also be reduced and used to synthesize cys by CysM and NrdH/Grx1. H_2_S catabolism: H_2_S binds to sulfur quinone oxidoreductase (SQR), and goes through a series of sequential reactions, including oxidation of sulfide to polysulfide by membrane-bound SQR, formation of GSSH from reaction involving with rhodanese (Rhd), oxidation of the sulfane sulfur in GSSH to SO_3_^2−^ catalyzed by persulfide dioxygenase (PDO), formation of S_2_O_3_^2−^ from spontaneous reaction between polysulfide and SO_3_^2−^, and formation of SO_4_^2−^ either spontaneously or catalyzed by various enzymes, transferring two electrons via quinone into the electron transport chain.

**Figure 3 antioxidants-11-02487-f003:**
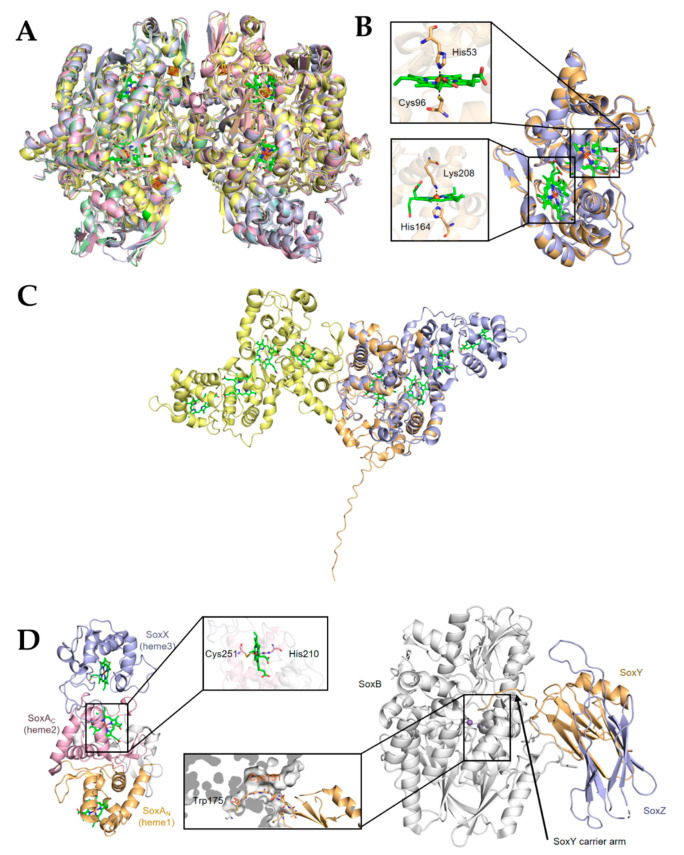
Structure of representative enzymes involved in transformation of sulfur compounds in bacteria. (**A**) Overall structure of the Dsr A2B2 heterotetramer of *Desulfovibrio vulgaris* (PDB ID: 2V4J), green; *Archaeoglobus fulgidus* (PDB ID: 3MM5), yellow; *Desulfovibrio gigas* (PDB ID: 3OR1), pink; and *Desulfoicrobrium norvegicum* (PDB ID: 2XSJ), blue. Heme ligands are shown in ball-stick model. (**B**) Structure of TsdA of *Marichromatium purpuratum* (PDB ID: 5LO9; violet) and of *Allochromatium vinosum* (PDB ID: 4WQ7; orange). Heme ligands are shown in ball-stick model. Expanded regions show the ligands to hemes in active centers. (**C**) TsdA of *Campylobacter jejuni* (orange, prepared from AlaphaFold database [90,91]) is aligned and superimposed onto TsdAB of *M. purpuratum* (PDB ID: 5LO9), which are in violet and yellow, respectively. Heme ligands are shown in ball-stick model. Expanded region shows the ligands to heme 2. (**D**) Structure of the SoxXA (PDB ID: 2C1D) (left) and SoxYZ-B (PDB ID: 4UWQ) (right) complexes. Heme ligands are shown in ball-stick model. Expanded region shows the active site positioning at the substrate channel of SoxB.

**Figure 4 antioxidants-11-02487-f004:**
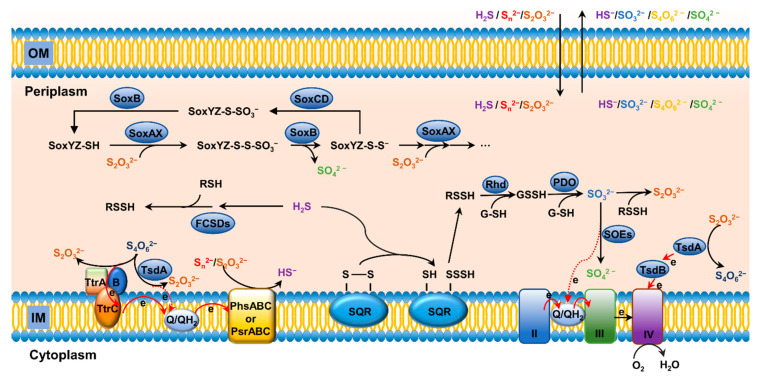
Pathways for bacterial sulfur transformation in the periplasm. Oxidation of S^2−^ and S^0^: Flavocyt *c* sulfide dehydrogenases (FCSDs) can oxidize H_2_S to the final product polysulfide. In dissimilatory sulfur-oxidizing bacteria, Rhds and PDOs are located in the periplasm, and the oxidation of H_2_S is catalyzed by SQRs, which consistently expose the reaction to the periplasm space. Oxidation of S^0^ and S^2+^: Unconjugated SoxYZ is catalyzed by SoxAX with S_2_O_3_^2−^, generating SoxYZ-S-S-SO_3_^−^, which is subsequently converted to SoxYZ-S-S^−^, releasing one molecular of SO_4_^2−^ under the catalysis of SoxB. Reduction of S^2+^: TtrABC and TsdAB are responsible for interconversion of S_4_O_6_^2−^ and S_2_O_3_^2−^. Reduction of S^0^: PhsABC and PsrABC are supposed to participate in the reduction of S^0^ (S_n_^2−^) and S^0^ (S_2_O_3_^2−^) to HS^−^.

**Figure 5 antioxidants-11-02487-f005:**
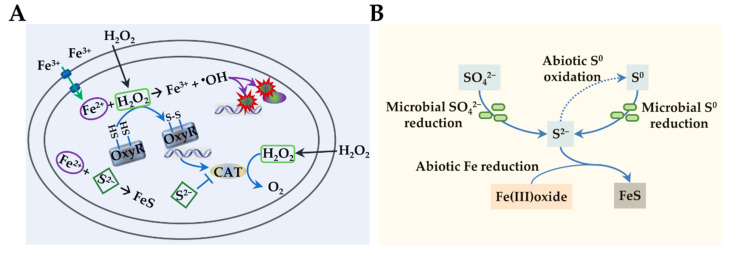
Physiological impacts of hydrogen sulfide. (**A**) H_2_S in oxidative stress. Upon oxidative stress, macromolecules such as DNA and proteins are damaged primarily by ^•^OH, which is generated from the interaction of Fe^2+^ and H_2_O_2_. H_2_S is a strong inhibitor of hemoproteins, catalase (CAT) in particular. With CAT inhibited, cells are unable to promptly decompose H_2_O_2_, leading to increased sensitivity to H_2_O_2_ killing. However, the prolonged presence of H_2_O_2_ activates OxyR, the master regulator in response to oxidative stress, which in turn induces expression of genes under its control, including CAT. As a result, cells gain an increased resistance against H_2_O_2_. In addition, it has been suggested that H_2_S from endogenous and exogenous sources may offer protection against oxidative stress by sequestering free Fe^2+^ intracellularly. (**B**) H_2_S in metal reduction. Microbial reduction of SO_4_^2−^ and elemental S^0^ to S^2−^, which catalyzes abiotic reduction of Fe^3+^ of Fe(III)oxide and forms FeS with the resulting Fe^2+^.

**Figure 6 antioxidants-11-02487-f006:**
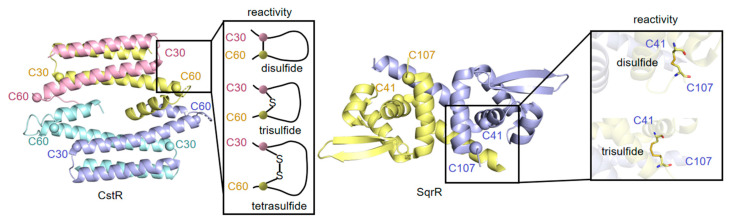
Structures and responding mechanism of CstR to RSS. Left, CstR (PDB ID: 7MQ2), which exists as homotetramer. Right, SqrR (PDB: 6O8N), which exists as homodimer. Expanded regions show the mechanism of activation. Both regulators use multiple cys thiols for sensing RSS by forming di-, tri-, and tetra-sulfide bonds.

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
