# Peer review of "Generation and Physiology of Hydrogen Sulfide and Reactive Sulfur Species in Bacteria"

_antioxidants, 2022, doi:10.3390/antiox11122487_

Round 1
Reviewer 1 Report
It is an interesting review article on H2S and reactive sulfur species in bacteria. There are a few comments.
1. 4.6 Oxidation of S4+. Sulfite has recently been reported in mammals to be protective against oxidative stress induced by high concentrations of glutamate called oxytosis (Br J Pharm 176, 571-582, 2019), which is recently suggested to be equivalent to ferroptosis (Cell Chem Biol 27, 1456-1471, 2020).
2. 5.2 Role of H2S in antibiotics. The interaction of H2S and NO produces H2Sn (Sci Rep 7: 45995, 2017).
Author Response
- 6 Oxidation of S4+. Sulfite has recently been reported in mammals to be protective against oxidative stress induced by high concentrations of glutamate called oxytosis (Br J Pharm 176, 571-582, 2019), which is recently suggested to be equivalent to ferroptosis (Cell Chem Biol 27, 1456-1471, 2020).
Kimura, Y.; Shibuya, N.; Kimura, H., Sulfite protects neurons from oxidative stress. Br J Pharmacol 2019, 176, (4), 571-582.
Maher, P.; Currais, A.; Schubert, D., Using the Oxytosis/Ferroptosis Pathway to Understand and Treat Age-Associated Neurodegenerative Diseases. Cell Chem Biol 2020, 27, (12), 1456-1471.
We mentioned these.
- 2 Role of H2S in antibiotics. The interaction of H2S and NO produces H2Sn (Sci Rep 7: 45995, 2017)
Miyamoto, R.; Koike, S.; Takano, Y.; Shibuya, N.; Kimura, Y.; Hanaoka, K.; Urano, Y.; Ogasawara, Y.; Kimura, H., Polysulfides (H(2)S(n)) produced from the interaction of hydrogen sulfide (H(2)S) and nitric oxide (NO) activate TRPA1 channels. Sci Rep 2017, 7, (45995).
We added this.
Reviewer 2 Report
In this manuscript, Gao et al., presented a global overview of Hydrogen sulfide and reactive sulfur species in bacteria. The manuscript is well written, offering the readership the broad range of highly relevant physiological examples of the roles of H2S and RSS in bacteria.
The manuscript is supported with the updated and the most recent insights which reflect the understanding of H2S and its corresponding RSS in bacteria.
This material offers the valuable asset and it will strongly contribute to the scientific field of H2S and RSS.
46. Therefore, “reactive” sulfur atoms can be found in compounds having sulfur of higher valence, such as sulfite 47 (SO32−, +4) and sulfate (SO42−, +6) which have interchalcogen bonds, and thiosulfate (S2O32−, 48 -1/+5 or 0/+4) in which sulfur atoms are catenated as in polysulfides (Figure 1).
The term “reactive” in this sentence can be potentially confusing.
60. Formation and metabolism of RSSs have been “well studied” in eukaryotes, especially mammals [9,11].
Well studied is slightly wrong term in particularly when it is associated with mammals. Many recent publications referring the formation of RSS are performed using and demonstrating not so well characterized and defined analytes (despite the application of corresponding cutting edge technologies) referring to RSS and its interpretations in mammals require additional thermodynamic and chemistry-based investigation. Authors are only advised not to use the linguistic construct such is: “well-studied” in the sentence containing formation of RSS in mammals.
111. It would be worth mentioning and incorporating the term “ Boring billon” and its connotation in this chapter.
296. It may be interesting to mention the application of a relatively simple tetrathionate salt-based molecule that exert the protective effect against ischemia-reperfusion ROS-derived injuries in mammals (DOI: /10.1371/journal.pmed.1002310).
489. In proteins containing a disulfide bond, a nucleophilic attach by the sulfide anion would cause the persulfide modification.
The sentence above is not clear and it is missing the appropriate reference.
Author Response
- Therefore, “reactive” sulfur atoms can be found in compounds having sulfur of higher valence, such as sulfite 47(SO32−, +4) and sulfate (SO42−, +6) which have interchalcogen bonds, and thiosulfate (S2O32−, 48-1/+5 or 0/+4) in which sulfur atoms are catenated as in polysulfides (Figure 1).
The term “reactive” in this sentence can be potentially confusing.
We added quotation marks to imply that this atom may not be the same as the typical ones.
- Formation and metabolism of RSSs have been “well studied” in eukaryotes, especially mammals [9,11].
Well studied is slightly wrong term in particularly when it is associated with mammals. Many recent publications referring the formation of RSS are performed using and demonstrating not so well characterized and defined analytes (despite the application of corresponding cutting edge technologies) referring to RSS and its interpretations in mammals require additional thermodynamic and chemistry-based investigation. Authors are only advised not to use the linguistic construct such is: “well-studied” in the sentence containing formation of RSS in mammals.
We changed to ‘intensively’.
- It would be worth mentioning and incorporating the term “ Boring billon” and its connotation in this chapter.
We added this.
Mukherjee, I.; Large, R. R.; Corkrey, R.; Danyushevsky, L. V., The Boring Billion, a slingshot for Complex Life on Earth. Scientific Reports 2018, 8, (1), 4432.
- It may be interesting to mention the application of a relatively simple tetrathionate salt-based molecule that exert the protective effect against ischemia-reperfusion ROS-derived injuries in mammals (DOI: /10.1371/journal.pmed.1002310).
Dyson, A.; Dal-Pizzol, F.; Sabbatini, G.; Lach, A. B.; Galfo, F.; Dos Santos Cardoso, J.; Pescador Mendonça, B.; Hargreaves, I.; Bollen Pinto, B.; Bromage, D. I.; Martin, J. F.; Moore, K. P.; Feelisch, M.; Singer, M., Ammonium tetrathiomolybdate following ischemia/reperfusion injury: Chemistry, pharmacology, and impact of a new class of sulfide donor in preclinical injury models. PLoS Med 2017, 14, (7).
We added this.
- In proteins containing a disulfide bond, a nucleophilic attach by the sulfide anion would cause the persulfide modification.
The sentence above is not clear and it is missing the appropriate reference.
We revised the sentence and provided a reference.
Reviewer 3 Report
It is a very well written manuscript. The topic is covered in depth and extensively with the applications of sulfur producing bacteria.
However, there are some minor suggestions for authors.
Though work is novel but it is important to demonstrate the novelty and importance of the work in abstract section more clearly.
Regarding the figure. I would like to know are these figures original or copied from literature. Pls write the sources of figure. As I can see that most of the Fig are taken from literature. Authors must provide some original/self-drawn fig.
Authors should provide future prospects of current MS.
Authors should provide a table of different sulfur generating bacteria and their application.
Section 2. Influence of sulfur on the origin of life: Authors should describe the role of production of H2S in cancer and neurological disorders in human.
https://www.mdpi.com/1420-3049/27/11/3389
write 4 to 5 sentences about Physiological and Pathological Roles of H2S in human.
Authors must check the uniformity of references.
There are some grammar and typo error.
Author Response
Though work is novel but it is important to demonstrate the novelty and importance of the work in abstract section more clearly.
Due to the word limitation, the abstract can only cover the main elements discussed in the text. For review articles, novelty and importance are reflected by papers they discuss.
Regarding the figure. I would like to know are these figures original or copied from literature. Pls write the sources of figure. As I can see that most of the Fig are taken from literature. Authors must provide some original/self-drawn fig.
All figures are original/self-drawn.
Authors should provide future prospects of current MS.
Concluding remarks is sort of future prospects.
Authors should provide a table of different sulfur generating bacteria and their application.
This review focuses on mechanisms underlying RSS biology in bacteria. Such a table, which could be extremely long, would not add information that is critically associated with our focus.
Section 2. Influence of sulfur on the origin of life: Authors should describe the role of production of H2S in cancer and neurological disorders in human.
https://www.mdpi.com/1420-3049/27/11/3389
write 4 to 5 sentences about Physiological and Pathological Roles of H2S in human.
We believe that this reference is valuable so we cite it in the text. But physiological and Pathological Roles of H2S in human is not critically relevant to the focus of this review.
Authors must check the uniformity of references.
We checked and revised.
There are some grammar and typo error.
We checked and corrected.
Reviewer 4 Report
1) The concept of the role of bacterial H2S as a protective factor against host immune response should be covered. Here are some examples: Infect Immun. 2018 Dec 19;87(1):e00272-18; Proc Natl Acad Sci U S A. 2020 Mar 24;117(12):6663-6674. 2) Also some papers are cited twice, eg the Nudler science paper. Please check referencing.Author Response
- The concept of the role of bacterial H2S as a protective factor against host immune response should be covered. Here are some examples: Infect Immun. 2018 Dec 19;87(1):e00272-18; Proc Natl Acad Sci U S A. 2020 Mar 24;117(12):6663-6674.
We added a paragraph for covering this concept. Thanks.
Toliver-Kinsky, T.; Cui, W.; Törö, G.; Lee, S. J.; Shatalin, K.; Nudler, E.; Szabo, C., H(2)S, a Bacterial Defense Mechanism against the Host Immune Response. Infect Immun 2018, 87, (1), 00272-18.
Rahman, M. A.; Cumming, B. M.; Addicott, K. W.; Pacl, H. T.; Russell, S. L.; Nargan, K.; Naidoo, T.; Ramdial, P. K.; Adamson, J. H.; Wang, R.; Steyn, A. J. C., Hydrogen sulfide dysregulates the immune response by suppressing central carbon metabolism to promote tuberculosis. Proc Natl Acad Sci U S A 2020, 117, (12), 6663-6674.
Shatalin, K.; Nuthanakanti, A.; Kaushik, A.; Shishov, D.; Peselis, A.; Shamovsky, I.; Pani, B.; Lechpammer, M.; Vasilyev, N.; Shatalina, E.; Rebatchouk, D.; Mironov, A.; Fedichev, P.; Serganov, A.; Nudler, E., Inhibitors of bacterial H<sub>2</sub>S biogenesis targeting antibiotic resistance and tolerance. Science 2021, 372, (6547), 1169-1175.
2) Also some papers are cited twice, eg the Nudler science paper. Please check referencing.
We corrected errors in references.